# Lion’s Mane Mushroom (*Hericium erinaceus*): A Neuroprotective Fungus with Antioxidant, Anti-Inflammatory, and Antimicrobial Potential—A Narrative Review

**DOI:** 10.3390/nu17081307

**Published:** 2025-04-09

**Authors:** Alex Graça Contato, Carlos Adam Conte-Junior

**Affiliations:** 1Analytical and Molecular Laboratorial Center (CLAn), Institute of Chemistry (IQ), Federal University of Rio de Janeiro (UFRJ), Cidade Universitária, Rio de Janeiro 21941-909, RJ, Brazil; conte@iq.ufrj.br; 2Center for Food Analysis (NAL), Technological Development Support Laboratory (LADETEC), Federal University of Rio de Janeiro (UFRJ), Cidade Universitária, Rio de Janeiro 21941-598, RJ, Brazil; 3Laboratory of Advanced Analysis in Biochemistry and Molecular Biology (LAABBM), Department of Biochemistry, Federal University of Rio de Janeiro (UFRJ), Cidade Universitária, Rio de Janeiro 21941-909, RJ, Brazil; 4Graduate Program in Biochemistry (PPGBq), Institute of Chemistry (IQ), Federal University of Rio de Janeiro (UFRJ), Cidade Universitária, Rio de Janeiro 21941-909, RJ, Brazil; 5Graduate Program in Food Science (PPGCAL), Institute of Chemistry (IQ), Federal University of Rio de Janeiro (UFRJ), Cidade Universitária, Rio de Janeiro 21941-909, RJ, Brazil; 6Graduate Program in Veterinary Hygiene (PPGHV), Faculty of Veterinary Medicine, Fluminense Federal University (UFF), Niterói 24220-000, RJ, Brazil; 7Graduate Program in Chemistry (PGQu), Institute of Chemistry (IQ), Federal University of Rio de Janeiro (UFRJ), Cidade Universitária, Rio de Janeiro 21941-909, RJ, Brazil

**Keywords:** bioactive compounds, monkey head mushroom, natural antimicrobials, polysaccharides, pom-pom mushroom, terpenoids

## Abstract

*Hericium erinaceus*, commonly known as lion’s mane mushroom, has gained increasing scientific interest due to its rich composition of bioactive compounds and diverse health-promoting properties. This narrative review provides a comprehensive overview of the nutritional and therapeutic potential of *H. erinaceus*, with a particular focus on its anti-inflammatory, antioxidant, and antimicrobial activities. A structured literature search was performed using databases such as PubMed, Scopus, Science Direct, Web of Science, Science Direct, and Google Scholar. Studies published in the last two decades focusing on *H. erinaceus*’ bioactive compounds were included. The chemical composition of *H. erinaceus* includes polysaccharides, terpenoids (hericenones and erinacines), and phenolic compounds, which exhibit potent antioxidant effects by scavenging reactive oxygen species (ROS) and inducing endogenous antioxidant enzymes. Additionally, *H. erinaceus* shows promising antimicrobial activity against bacterial and fungal pathogens, with potential applications in combating antibiotic-resistant infections. The mushroom’s capacity to stimulate nerve growth factor (NGF) synthesis has highlighted its potential in preventing and managing neurodegenerative diseases, such as Alzheimer’s and Parkinson’s. Advances in biotechnological methods, including optimized cultivation techniques and novel extraction methods, may further enhance the bioavailability and pharmacological effects of *H. erinaceus*. Despite promising findings, clinical validation remains limited. Future research should prioritize large-scale clinical trials, the standardization of extraction methods, and the elucidation of pharmacokinetics to facilitate its integration into evidence-based medicine. The potential of *H. erinaceus* as a functional food, nutraceutical, and adjunct therapeutic agent highlights the need for interdisciplinary collaboration between researchers, clinicians, and regulatory bodies.

## 1. Introduction

The growing interest in natural products with anti-inflammatory, antioxidant, and antimicrobial properties is driven by scientific advancements and shifting consumer preferences toward safer and more sustainable health solutions [1,2,3,4]. Chronic inflammation and oxidative stress are underlying factors in a wide range of diseases, including cardiovascular disorders, diabetes, neurodegenerative illnesses, and cancer [5,6,7]. Conventional treatments often rely on synthetic drugs, which, despite their efficacy, can lead to significant side effects and long-term health risks [8]. As a result, researchers and healthcare professionals are increasingly turning to bioactive compounds from natural sources as potential alternatives or complementary therapies [9]. These compounds, derived from plants, fungi, and marine organisms, have demonstrated the ability to regulate inflammatory pathways, neutralize reactive oxygen species (ROS), and combat microbial infections, offering promising prospects for disease prevention and treatment [10,11,12].

In parallel, the rise in antibiotic-resistant pathogens has intensified the global search for novel antimicrobial agents [7,13]. The overuse and misuse of antibiotics have led to the emergence of multidrug-resistant bacteria, posing a significant threat to public health [14]. Natural products, particularly those from medicinal plants and fungi, have shown potent antimicrobial activity through diverse mechanisms, such as disrupting bacterial membranes, inhibiting biofilm formation, and modulating microbial metabolism [15,16,17]. As scientific interest in these bioactive molecules grows, there is a rising demand for functional foods, nutraceuticals, and novel therapeutics that harness the health benefits of natural compounds, where mushrooms and their extracts can be a promising alternative [18,19,20].

Mushrooms have long been recognized for their nutritional and therapeutic value [21,22]. In recent years, they have gained increasing attention in modern scientific research due to their rich composition of bioactive compounds and potential health benefits [23,24]. They contain unique secondary metabolites such as polysaccharides (notably β-glucans), terpenoids, polyphenols, and peptides, which contribute to their immunomodulatory, antioxidant, anti-inflammatory, and antimicrobial properties [21]. From a nutritional perspective, mushrooms are a valuable source of essential nutrients, including proteins, dietary fiber, vitamins (such as complex B and vitamin D precursors), and minerals like selenium, zinc, and potassium, sources of antioxidant protection, and immune support [25]. Their low-calorie content and high nutrient density make them an excellent addition to a balanced diet, particularly for individuals seeking plant-based sources of protein and bioactive compounds [21]. Furthermore, many mushrooms contain prebiotic fibers that support gut health by promoting the growth of beneficial microbiota, contributing to improved digestion and immune function [26,27].

Given their impressive range of nutritional and health benefits, mushrooms are becoming key ingredients in functional foods and nutraceuticals [21]. Nowadays, they are being incorporated into powders, capsules, teas, and even plant-based meat alternatives, catering to consumers who seek natural solutions for overall wellness and disease prevention [21,28].

Among the mushrooms, *Hericium erinaceus*, commonly known as lion’s mane mushroom, has been the subject of extensive research in recent years due to its promising bioactive compounds [29]. It is recognized as a purported benefit in enhancing cognitive function and supporting gastrointestinal health [30]. This unique mushroom is characterized by its distinct appearance, with long, white spines resembling a lion’s mane [31]. Studies have highlighted its potential in promoting neurogenesis, improving memory and concentration, and protecting against neurodegenerative diseases [32]. *H. erinaceus* contains a variety of bioactive compounds, which include polysaccharides, terpenoids, and phenolic compounds. Polysaccharides, particularly β-glucans, are known for their immunomodulatory and neuroprotective effects. Terpenoids, such as hericenones and erinacines, have been shown to stimulate nerve growth factor (NGF) synthesis, promoting neuronal growth and repair. Additionally, phenolic compounds present in *H. erinaceus* exhibit strong antioxidant properties, helping to mitigate oxidative stress and inflammation [29]. These properties make *H. erinaceus* an attractive candidate for applications in functional foods, nutraceuticals, and even pharmaceutical development [33,34,35].

This narrative review aims to explore the bioactive compounds found in *H. erinaceus*, particularly its polysaccharides, terpenoids, and phenolic compounds, with a special emphasis on their neuroprotective properties. By highlighting the ability of *H. erinaceus* to stimulate NGF synthesis, combat neuroinflammation, and protect against oxidative stress, this review underscores its therapeutic potential in the prevention and management of neurodegenerative diseases. Furthermore, this discussion will extend to its broader health benefits, reinforcing its relevance as a functional food and natural neuroprotective agent in modern healthcare. Specifically, this narrative review seeks to provide a comprehensive understanding of the mechanisms through which *H. erinaceus* exerts its effects, summarize current scientific findings, and identify potential gaps in knowledge that warrant further research.

## 2. Methods

This narrative review was performed following three steps: conducting the search, reviewing abstracts and full texts, and discussing the results. For this, the PubMed, Scopus, Science Direct, Web of Science, Science Direct, and Google Scholar databases were searched to identify relevant studies, according to the development of the review. The final search was conducted in March 2025 and included international English-language articles, online reports, and electronic books. The keyword “*Hericium erinaceus*” was used in combination with other terms such as characteristics, habitat, chemical composition, cultivation methods, anti-inflammatory activity, clinical trials, bioavailability, blood–brain barrier penetration, antioxidant activity, antimicrobial activity, calcium binding activity, nutritional and therapeutic applications, challenges, or regulation. After the complete search, the abstracts were read to ensure that they addressed the topic of interest. All duplicates were removed, and the abstracts of the remaining articles were reviewed to ensure that they addressed the inclusion criteria of the review. The eligible criteria were studies that analyzed *Hericium erinaceus* in combination with the other terms mentioned above. Therefore, the studies of interest were summarized and synthesized to integrate the narrative review. Since it is a narrative review, it was not necessary to document the literature search on specific platforms [36]

## 3. Characteristics, Habitat, and Chemical Composition of *H. erinaceus*

### 3.1. Taxonomy and Morphology

*H. erinaceus* belongs to the Kingdom Fungi, Phylum Basidiomycota, Class Agaricomycetes, Order Russulales, and Family Hericiaceae (Figure 1) [37]. The *Hericium* genus includes several species, such as *Hericium coralloides* and *Hericium americanum*, but *H. erinaceus* stands out due to its medicinal properties and distinctive morphology [38]. The species name was assigned by Christian Hendrik Persoon in the 19th century [39]. It is commonly known as “lion’s mane”, “monkey head”, or “pom-pom mushroom” due to its unique appearance [32].

*H. erinaceus* has a globular or semi-spherical fruiting body, and is white to cream-colored when young, turning yellowish or brownish with maturation. Its most distinctive feature is the presence of long, hanging spines measuring 1 to 5 cm in length, covering the entire surface of the basidiocarp. Unlike more common edible mushrooms, which have a cap-and-stem structure, *H. erinaceus* lacks distinct morphological differentiation, instead growing in a compact form with densely arranged spines [40].

The mycelium of *H. erinaceus* is white and robust [40], thriving on lignocellulosic substrates [41,42]. Its reproductive system relies on the production of basidiospores, which are ellipsoid to cylindrical, smooth, and measure 5–7 × 4–5 µm. During sporulation, a cloud of white spores can often be seen around the mature mushroom [40].

Although *H. erinaceus* shares some characteristics with other species in the *Hericium* genus, it can be distinguished by its more compact shape and the uniform arrangement of its spines [40]. In contrast, *H. coralloides* have a more branched, coral-like structure with shorter spines distributed in several directions, while *H. americanum* has an intermediate morphology, with branching similar to that of *H. coralloides* but with longer spines, resembling *H. erinaceus* [43].

### 3.2. Habitat and Cultivation Methods of H. erinaceus

*H. erinaceus* is a saprotrophic and weak parasitic fungus [38] that primarily colonizes hardwood trees, such as oak (*Quercus* spp.), beech (*Fagus* spp.), maple (*Acer* spp.), walnut (*Juglans* spp.), and birch (*Betula* spp.) [41,42]. It typically grows on dead or dying trees, essential in decomposing lignocellulosic material [44]. This species prefers temperate forests in North America, Europe, and Asia, thriving in regions with high humidity and moderate temperatures, but it has already expanded to the most diverse regions of the planet and is marketed globally [21,45].

In the wild, *H. erinaceus* is commonly found during late summer and autumn, when environmental conditions favor its fruiting [46]. It tends to grow at elevated positions on tree trunks, making it harder to spot and collect compared to ground-growing mushrooms [47].

As a white-rot fungus, *H. erinaceus* degrades lignin more efficiently than cellulose, breaking down complex organic matter and recycling nutrients into the ecosystem [44]. Its ability to colonize trees while they are still alive classifies it as a facultative parasite, meaning it can survive on both living and dead wood. This dual nature allows *H. erinaceus* to persist in forests for extended periods, contributing to biodiversity and forest regeneration [48].

Commercial cultivation has become essential to meet market needs with the rising demand for *H. erinaceus* in functional foods, nutraceuticals, and medicinal applications [49]. Unlike wild harvesting, controlled cultivation offers higher yields, improved quality, and year-round availability [50]. There are three main methods for cultivating *H. erinaceus*: log cultivation, supplemented sawdust blocks, and liquid culture fermentation [51]. A comparison of the characteristics and properties of the cultivation methods of *H. erinaceus* can be seen in Table 1.

### 3.3. General Chemical Composition of H. erinaceus

The lion’s mane mushroom is renowned for its diverse and bioactive chemical composition, which contributes to its wide range of health benefits [58]. Its key bioactive compounds include polysaccharides, terpenoids, phenolic compounds, and bioactive proteins, each playing a significant role in its antioxidant, anti-inflammatory, and neuroprotective properties [59,60]. Some examples of bioactive proteins found include lectins, carbohydrate-binding proteins which exhibit immunomodulatory and antimicrobial activities [61], and glucanases and chitinases, enzymes that degrade fungal polysaccharides and stimulate immune responses [62]. Oxidative enzymes involved in lignin degradation and detoxification like laccases and peroxidases have demonstrated antioxidant and antibacterial properties [63], while ribosome-inactivating proteins (RIPs) can inhibit protein synthesis in target cells, potentially exhibiting cytotoxic effects against tumors or pathogens [64]. Additionally, hydrophobins are surface-active proteins that play roles in fungal adhesion and biofilm formation, with emerging biomedical applications [65].

Polysaccharides, particularly β-glucans, are among the most well-studied bioactive compounds in *H. erinaceus* [60]. These complex carbohydrates are known for their immunomodulatory, antimicrobial, and antitumor effects [66,67]. β-glucans can stimulate the immune system by activating macrophages, natural killer (NK) cells, and T lymphocytes, enhancing the body’s ability to fight infections and even cancer cells [68]. Other polysaccharides found in *H. erinaceus* include heteropolysaccharides composed of glucose, mannose, galactose, and arabinose [69]. These compounds have demonstrated the ability to reduce oxidative stress, regulate blood sugar levels, and improve gut microbiota by acting as prebiotic fibers [70,71].

Terpenoids are another significant class of bioactive compounds in *H. erinaceus*, with two key groups: hericenones (found in the fruiting body) and erinacines (found in the mycelium) [72]. These compounds are mainly known for their neuroregenerative and neuroprotective properties [73], as they stimulate the synthesis of nerve growth factor (NGF), an essential protein for the growth, maintenance, and survival of neurons, making them particularly relevant for neurodegenerative diseases like Alzheimer’s (AD) and Parkinson’s (PD) [72]. Among them, erinacines have been extensively studied for their ability to cross the blood–brain barrier and exert potent neuroprotective effects [30,72]. Hericenones, on the other hand, have demonstrated potential in cognitive enhancement and memory improvement, making *H. erinaceus* a promising candidate for nootropic applications [72].

In addition to their effects on neurogenesis, both hericenones and erinacines exhibit anti-inflammatory and antimicrobial properties, making them valuable in reducing chronic inflammation and combating bacterial infections [74]. By modulating key inflammatory pathways, such as NF-κB and COX-2 (cyclooxygenase-2) inhibition, they contribute to reducing neuroinflammation, chronic systemic inflammation, and associated diseases, including autoimmune disorders [72]. Furthermore, certain erinacines have shown antimicrobial activity, particularly against *Helicobacter pylori*, suggesting a role in gut health and gastric ulcer prevention [49]. Table 2 shows a list of main hericenones and erinacines found in *H. erinaceus*.

Moreover, *H. erinaceus* contains ergothioneine, a histidine-derived amino acid with potent antioxidant properties [99]. Ergothioneine has garnered increasing interest due to its ability to neutralize ROS and reduce oxidative stress in neuronal cells. Unlike many dietary antioxidants, ergothioneine is actively transported into cells via the OCTN1 transporter, granting it distinct bioavailability [100]. Studies suggest that its neuroprotective action may have implications in preventing neurodegenerative diseases such AD and PD, where oxidative stress plays a central role [78,99].

The biological activity of *H. erinaceus* is directly related to the concentration of its active compounds. Studies indicate that the levels of hericenones and erinacines vary depending on the cultivation substrate and the fungal developmental stage [101]. For instance, hericenones extracted from the fruiting body can be present at concentrations ranging from <20 to 500 µg/g of dry weight, whereas erinacines, found in the mycelium, can reach concentrations ~150 µg/g [90]. Additionally, ergothioneine in *H. erinaceus* has been detected at levels between 0.34 and 1.30 mg/g, depending on cultivation conditions [78,100]. Accurate quantification of these compounds is essential for understanding their bioavailability and therapeutic efficacy.

Phenolic compounds, including gallic acid, caffeic acid, and *p*-coumaric acid, are also present in *H. erinaceus* and contribute to its strong antioxidant capacity [33]. These compounds act by scavenging reactive oxygen species and inducing endogenous antioxidant enzymes such as superoxide dismutase (SOD) and glutathione peroxidase (GPx) [9]. By mitigating oxidative stress, phenolic compounds help protect against cellular damage, aging-related diseases, and inflammatory conditions [5,7].

In addition, *H. erinaceus* is also a rich source of essential nutrients, including proteins, dietary fiber, vitamins like complex B (B1, B2, B3, B5, B6) and vitamin D precursors, and minerals, which contribute to several physiological processes, including antioxidant defense and nerve function [102].

The chemical structures of main constituents of *H. erinaceus*, including hericenones, erinacines, and other relevant molecules, are shown in Figure 2.

Compared to other well-known medicinal mushrooms, *H. erinaceus* stands out primarily for its neuroprotective properties, largely attributed to its unique erinacines and hericenones [72,78,93]. *Ganoderma lucidum* (reishi), for instance, is widely recognized for its strong immunomodulatory and anti-cancer effects due to its high content of triterpenoids and polysaccharides [103]. *Cordyceps sinensis* (now *Ophiocordyceps sinensis*), another medicinal mushroom, is known for its energy-boosting and anti-fatigue properties, largely attributed to cordycepin and adenosine derivatives, which enhance mitochondrial function [104]. Meanwhile, *Lentinula edodes* (shiitake) is particularly rich in lentinan, a β-glucan with immunomodulatory and anti-cancer properties [105]. Another notable species, *Phellinus linteus*, exhibits potent anti-tumor and anti-inflammatory activities through its unique polyphenolic compounds [22]. Unlike these mushrooms, *H. erinaceus* remains unparalleled in its ability to stimulate NGF synthesis, making it a promising candidate for neurodegenerative disease treatment and cognitive enhancement [72,78]. This distinct biochemical profile underscores its unique therapeutic niche among medicinal fungi.

## 4. Biological Properties and Mechanisms of Action

### 4.1. Anti-Inflammatory Activity

Inflammation is a complex biological response to harmful stimuli, including pathogens, damaged cells, or irritants [106]. While acute inflammation is essential for healing and defense [107], chronic inflammation is associated with several diseases, including cancer, cardiovascular diseases, diabetes, and neurodegenerative disorders [108]. The anti-inflammatory effects of *H. erinaceus* are attributed to several bioactive compounds that interact with key inflammatory pathways, regulating cytokine production, oxidative stress, and gut microbiota balance [109].

The bioactive compounds in *H. erinaceus* exert anti-inflammatory effects through multiple mechanisms, including modulation of key signaling pathways and inflammatory mediators [110].

The NF-κB signaling pathway plays a central role in inflammation by regulating the transcription of pro-inflammatory genes [111]. Activation of NF-κB leads to the increased production of cytokines such as TNF-α, IL-6, and IL-1β [112]. Erinacines and hericenones inhibit the phosphorylation of IκBα [72], preventing NF-κB activation [113] and nuclear translocation (Figure 3A) [87]. Polysaccharides suppress NF-κB signaling in macrophages, reducing the release of inflammatory mediators [114]. This inhibition of NF-κB signaling is particularly relevant in the context of neuroinflammation, as chronic activation of this pathway has been linked to the progression of AD and PD [115]. By suppressing neuroinflammation, *H. erinaceus* may contribute to slowing cognitive decline and protecting neuronal integrity in these disorders [97].

Additionally, *H. erinaceus* polysaccharides downregulate pro-inflammatory cytokines (IL-6, TNF-α, IL-1β) [109] while upregulating anti-inflammatory cytokines (IL-10) [116]. Erinacines have shown potential in neuroinflammation by suppressing glial cell activation and reducing IL-1β expression [85]. This is particularly relevant in Alzheimer’s disease, where excessive microglial activation contributes to amyloid-beta plaque formation and neuronal damage [117]. The ability of *H. erinaceus* to modulate glial function suggests a protective role against neurodegenerative damage [78].

*H. erinaceus* also acts in the inhibition of COX-2 (cyclooxygenase-2) and iNOS (inducible nitric oxide synthase) (Figure 3B) [102], where hericenones inhibit COX-2, reducing prostaglandin E2 (PGE2) synthesis, which plays an essential role in inflammation. *H. erinaceus* extracts also suppress iNOS expression, leading to reduced nitric oxide (NO) production, which is associated with chronic inflammation [118]. Elevated NO levels have been linked to oxidative stress and mitochondrial dysfunction in Parkinson’s disease [119].

The antioxidant properties of *H. erinaceus* contribute to its anti-inflammatory effects by activating the Nrf2 (nuclear factor erythroid 2-related factor 2) pathway, which enhances the expression of antioxidant enzymes like superoxide dismutase (SOD) and glutathione peroxidase (GPx) (Figure 3C) [120]. This antioxidant action is critical in neurodegenerative conditions such as AD and PD, where oxidative stress accelerates neuronal damage [121]. *H. erinaceus* polysaccharides also act as prebiotics, leading to reduced lipopolysaccharide (LPS)-induced inflammation and improved gut barrier function [111].

Several cell-based studies have demonstrated the anti-inflammatory potential of *H. erinaceus* extracts [34,35,74,110,122]. Polysaccharides and phenolic compounds from *H. erinaceus* significantly reduced the LPS-induced production of TNF-α, IL-6, and nitric oxide in RAW 264.7 macrophages [123]. Lee et al. [96] show that erinacine A inhibited pro-inflammatory cytokine expression in BV-2 microglial cells, suggesting neuroprotective potential, and Yang et al. [124] demonstrate that polysaccharides reduced inflammation in Caco-2 cells by modulating NF-κB signaling.

Animal studies have confirmed the anti-inflammatory effects of *H. erinaceus* in several disease models [109,125,126,127]. In a mouse model of Alzheimer’s disease, erinacines reduced neuroinflammation, and suppressed IL-1β expression [128]. Ren et al. [109] showed that *H. erinaceus* polysaccharides alleviated dextran sulfate sodium (DSS)-induced colitis in mice by restoring gut microbiota balance and reducing pro-inflammatory cytokines. In high-fat-diet-induced obese mice, supplementation with *H. erinaceus* extracts reduced systemic inflammation and improved insulin sensitivity [129].

### 4.2. Clinical Trials

Although much of the current research on *H. erinaceus* is based on animal and in vitro studies, several clinical trials have explored its potential benefits in humans, particularly in neurodegenerative diseases, cognitive function, and gastrointestinal health [130,131,132,133,134].

One of the most significant clinical trials investigated the effects of *H. erinaceus* supplementation on cognitive function in 50- to 80-year-old Japanese men and women with mild cognitive impairment (MCI). In a randomized, double-blind, placebo-controlled study, subjects who consumed *H. erinaceus* extract for 16 weeks showed significant improvements in cognitive performance compared to the placebo group. Notably, these benefits declined after the discontinuation of supplementation, suggesting a need for sustained intake to maintain cognitive enhancements [130].

Another trial focused on the potential neuroprotective effects of *H. erinaceus* in patients with early-stage Alzheimer’s disease. Preliminary findings indicated that regular consumption of *H. erinaceus* improved memory recall and reduced neuropsychiatric symptoms, likely due to its ability to stimulate NGF production and mitigate neuroinflammation. While promising, these results highlight the need for larger-scale studies with longer follow-up periods to establish definitive clinical efficacy [131].

Beyond cognitive function, clinical trials have also examined the role of *H. erinaceus* in gastrointestinal health [132,133]. A study in patients with gastritis found that supplementation with *H. erinaceus* significantly reduced inflammation-related symptoms, improved mucosal healing, and modulated gut microbiota composition. These findings suggest potential applications in managing gastrointestinal disorders such as irritable bowel syndrome (IBS) and inflammatory bowel disease (IBD), where chronic inflammation plays a central role [133].

Additionally, *H. erinaceus* has been evaluated for its effects on mood disorders, with clinical evidence suggesting its potential to alleviate symptoms of anxiety and depression. In a small-scale study, participants who consumed *H. erinaceus* extract reported reduced levels of stress and improved mood regulation, potentially linked to its influence on neurotrophic factors and inflammation-related pathways in the brain [134].

Despite these encouraging findings, clinical research on *H. erinaceus* remains limited, with many studies featuring small sample sizes and short durations. Future trials should aim to include larger, more diverse populations, employ standardized extract formulations, and explore long-term safety and efficacy. Establishing robust clinical evidence will be essential for validating *H. erinaceus* as a functional food or therapeutic agent in neurodegenerative and inflammatory diseases.

### 4.3. Bioavailability and Blood–Brain Barrier Penetration

A critical challenge in translating neuroprotective compounds into effective therapies is their bioavailability and ability to cross the blood–brain barrier (BBB). *H. erinaceus* contains bioactive terpenoids, particularly erinacines, that have demonstrated the ability to penetrate the BBB, a key advantage over many other natural compounds [135]. Erinacine A, for example, has been shown to increase nerve growth factor (NGF) levels in the brain, promoting neurogenesis and neuronal survival [136].

However, the overall bioavailability of *H. erinaceus* compounds remains a subject of investigation. Factors such as digestion, metabolism, and systemic distribution influence how these compounds reach target tissues [135]. Erinacines, being lipophilic [137], exhibit better BBB permeability compared to hydrophilic polysaccharides like β-glucans, which primarily exert effects through immune modulation rather than direct neuroprotection [76,138].

Advancements in delivery systems, such as nanoparticle-based formulations and lipid carriers, could enhance the absorption and brain-targeting efficacy of *H. erinaceus* extracts [139,140]. Encapsulation techniques have been explored to improve the stability and controlled release of bioactive compounds, potentially increasing their therapeutic effects in neurodegenerative conditions [141].

Further research is needed to elucidate the pharmacokinetics of *H. erinaceus* compounds in humans, including their metabolism, half-life, and optimal dosing strategies for neuroprotection [135]. Understanding these aspects will be crucial for developing clinically relevant applications and maximizing the therapeutic potential of this medicinal mushroom in conditions such as AD and PD [78].

### 4.4. Antioxidant Activity

*H. erinaceus* has been extensively studied for its potent antioxidant properties, which are attributed to several bioactive metabolites, including hericenones, erinacines, and polyphenolic compounds (i.e., gallic, caffeic, p-coumaric acids) [25,142]. To assess the antioxidant potential of *H. erinaceus*, several biochemical assays can be employed like DPPH (2,2-diphenyl-1-picrylhydrazyl), ABTS (2,2′-azino-bis(3-ethylbenzothiazoline-6-sulfonic acid)), FRAP (Ferric Reducing Antioxidant Power), ORAC (Oxygen Radical Absorbance Capacity), and TBARS (Thiobarbituric Acid Reactive Substances) [33,84,143,144]. The key mechanisms are as follows:*Reactive Oxygen Species (ROS) Modulation:* The bioactive compounds in *H. erinaceus* scavenge ROS, reducing oxidative stress at the cellular level [145]. This action prevents oxidative damage to lipids, proteins, and DNA, reducing the risk of chronic diseases [146].*Induction of Antioxidant Enzymes: H. erinaceus* extracts have been reported to upregulate the activity of antioxidant enzymes, including superoxide dismutase (SOD—converts superoxide radicals into less harmful molecules); catalase (CAT—breaks down hydrogen peroxide into water and oxygen, reducing cellular toxicity); and glutathione peroxidase (GPx—protects cells from oxidative damage by reducing peroxides) [145].*Inhibition of Lipid Peroxidation:* Studies have demonstrated that *H. erinaceus* extracts prevent the peroxidation of lipids, reducing malondialdehyde (MDA) levels, a significant factor in the aging process and the development of cardiovascular diseases [129].

Many studies have reported the antioxidant properties of the lion’s mane, especially its extracts [60,147,148,149]. A study by Lew et al. [145] demonstrated that *H. erinaceus* extract enhanced neuronal survival by reducing oxidative stress in brain cells. The researchers attributed this effect to increased NGF levels and the activation of antioxidant defense mechanisms. Lu et al. [129] investigated the hepatoprotective effects of *H. erinaceus* in an oxidative stress-induced liver injury model. Their findings showed that treatment with mushroom extract significantly increased SOD, CAT, and GPx levels, protecting liver cells from oxidative damage. Jalani [150] conducted a study on the DPPH and ABTS radical-scavenging activity of *H. erinaceus* extracts. Their results indicated a high antioxidant capacity, comparable to that of known natural antioxidants such as vitamin C and tocopherols. Research by Roda et al. [78] found that supplementation with *H. erinaceus* extracts reduced oxidative stress markers and improved endothelial function in an animal model of hypertension. This suggests a potential role in cardiovascular disease prevention.

### 4.5. Antimicrobial Activity

The lion’s mane, a medicinal mushroom with a rich history in traditional medicine, has demonstrated significant antimicrobial potential against several bacterial and fungal pathogens [33,58,151]. Its bioactive compounds, including polysaccharides, terpenoids, and phenolic compounds, contribute to its antimicrobial activity through several mechanisms, such as disrupting microbial cell membranes, inhibiting biofilm formation, and modulating host immune responses [110,152,153], which are explored in more detail below.

*Cell Membrane Disruption:* Several bioactive compounds in *H. erinaceus*, particularly terpenoids and phenolic compounds, have been shown to interfere with bacterial and fungal cell membranes [29,74]. These compounds disrupt membrane integrity by altering lipid bilayer stability, leading to increased permeability, the leakage of intracellular contents, and eventual cell death. This mechanism is particularly relevant against Gram-positive bacteria, which have a thick peptidoglycan layer that is more susceptible to membrane-targeting agents [154].*Inhibition of Biofilm Formation:* Biofilms are protective structures formed by microbial communities that enhance resistance to antibiotics and immune responses [155]. Polysaccharides and terpenoids from *H. erinaceus* have demonstrated the ability to inhibit biofilm formation by interfering with quorum sensing pathways, the bacterial communication system that regulates biofilm development [93]. By preventing biofilm maturation, *H. erinaceus* compounds enhance the susceptibility of bacteria to antimicrobial agents and host immune defenses [155].*Enzyme Inhibition and Metabolic Disruption:* Phenolic compounds in *H. erinaceus* have been reported to inhibit key bacterial enzymes involved in cell wall synthesis, DNA replication, and energy metabolism [69,156]. For example, some erinacines and hericenones have been shown to interfere with bacterial ATP production, disrupting essential metabolic pathways and leading to growth inhibition [93,145].*Induction of Oxidative Stress:* Some bioactive compounds in *H. erinaceus* promote the generation of reactive oxygen species in microbial cells [157]. Excess ROS accumulation leads to oxidative damage to proteins, lipids, and DNA, ultimately resulting in cell death [7]. This mechanism is particularly effective against antibiotic-resistant bacteria, which often rely on antioxidant defense systems to survive in hostile environments [142].*Modulation of Host Immune Responses:* Polysaccharides, especially β-glucans, play a crucial role in enhancing the host’s immune response against infections [67]. These compounds stimulate macrophages, dendritic cells, and NK cells, boosting antimicrobial activity and facilitating the clearance of bacterial and fungal pathogens [110].

*H. erinaceus* exhibits potent activity against Gram-positive bacteria, particularly *Staphylococcus aureus* (including methicillin-resistant *S. aureus* [MRSA]) [158], *Bacillus subtilis* [159], and *Enterococcus faecalis* [160]. The activity against Gram-negative bacteria is generally lower, but some studies report effects on *H. pylori* [161], and *Pseudomonas aeruginosa* [162]. The antifungal properties of *H. erinaceus* have been demonstrated against *Candida albicans* [162] and *Aspergillus flavus* [163].

The antimicrobial activity of *H. erinaceus* varies depending on the specific bioactive compounds and target microorganisms. In general, its effects are predominantly bacteriostatic, meaning it inhibits bacterial growth rather than directly killing bacteria [164]. This contrasts many conventional antibiotics, such as β-lactams (e.g., penicillins and cephalosporins), which exhibit bactericidal activity by disrupting bacterial cell walls and causing lysis [165]. However, some studies have reported bactericidal effects of *H. erinaceus* extracts, particularly against Gram-positive bacteria like *S. aureus* [166] and *B. subtilis* [167]. These effects are likely due to membrane-disrupting terpenoids, which resemble the mechanism of antimicrobial peptides [159] and some lipopeptide antibiotics (e.g., daptomycin) [168]. The ability of *H. erinaceus* to disrupt bacterial membranes suggests a mode of action similar to that of polymyxins, which are effective against Gram-negative bacteria, but with a different structural basis [168].

When compared to well-known antibiotics, *H. erinaceus* polysaccharides function similarly to macrolides (e.g., erythromycin) by inhibiting bacterial protein synthesis and biofilm formation, leading to reduced bacterial virulence [169]. Phenolic compounds in *H. erinaceus* exhibit antioxidant and antimicrobial properties comparable to those of quinolones (e.g., ciprofloxacin), which target bacterial DNA replication [170], while terpenoids from *H. erinaceus* act similarly to lipopeptide antibiotics by disrupting bacterial membranes [171].

One of the most promising aspects of *H. erinaceus* as an antimicrobial agent is its potential to enhance the efficacy of existing antibiotics [154]. Several studies suggest that its bioactive compounds may act synergistically with conventional antibiotics, improving antimicrobial activity [74,164,172]. Specific polysaccharides and terpenoids in *H. erinaceus* may enhance cell wall permeability, making bacteria more susceptible to β-lactam antibiotics [173].

While *H. erinaceus* has shown promise as an antimicrobial agent, challenges such as bioavailability and extraction efficiency need to be addressed before it can be incorporated into mainstream pharmaceuticals [174]. Given its broad antimicrobial spectrum and potential to enhance antibiotic efficacy, *H. erinaceus* holds promise for several applications, such as its use as an adjuvant therapy to enhance conventional antibiotics and its potential application in topical antimicrobial agents for wound healing and skin infections [175]. Additionally, *H. erinaceus* extracts could also be used as natural preservatives to extend shelf life and prevent microbial contamination in food products [162]. Studies have shown that mushroom-derived bioactives can inhibit the growth of foodborne pathogens such as *Listeria monocytogenes* and *Salmonella* spp. without the need for synthetic preservatives [176]. However, optimization of extraction methods and stability studies under different storage conditions are necessary to ensure practical implementation [177]. Despite these promising applications, further research is needed to determine the most effective delivery methods, safety profiles, and regulatory pathways for integrating *H. erinaceus* into clinical and industrial products [178]. Large-scale clinical trials and toxicological studies will be essential to validate its use as a therapeutic or preservative agent [142].

### 4.6. Calcium Binding Activity and Other Functional Properties

In addition to its well-documented neuroprotective, antioxidant, anti-inflammatory, and antimicrobial properties, *H. erinaceus* exhibits other biochemical activities of potential therapeutic significance, including calcium-binding capacity [179]. Calcium plays an essential role in numerous physiological processes, such as neuronal excitability, synaptic plasticity, muscle contraction, and intracellular signaling [180]. Dysregulation of calcium homeostasis is strongly implicated in the pathogenesis of neurodegenerative diseases, including Alzheimer’s, Parkinson’s, and Huntington’s diseases [181].

Recent studies suggest that specific polysaccharides and proteins present in *H. erinaceus* may interact with calcium ions, influencing calcium-dependent signaling pathways [156,179]. These pathways are fundamental for neuronal survival and synaptic function, and their disruption has been associated with cognitive decline and neurodegeneration [182]. For instance, certain fungal polysaccharides have been shown to modulate calcium influx and efflux, potentially reducing excitotoxicity, a condition characterized by excessive calcium entry leading to neuronal damage and apoptosis [183].

Moreover, calcium-binding proteins play a vital role in modulating oxidative stress and inflammatory responses, two major contributors to neurodegenerative processes [184]. Excess-free calcium can activate enzymes such as calpains and phospholipases, which, when dysregulated, lead to protein degradation, membrane damage, and neuronal death [185].

Beyond calcium-binding, *H. erinaceus* also displays metal ion chelating activity, which may contribute to neuroprotection [186]. Metal ions such as iron (Fe^2+^), copper (Cu^2+^), and zinc (Zn^2+^) play essential roles in normal brain function but can become neurotoxic when dysregulated [187]. Excess accumulation of these metals has been implicated in oxidative stress and the aggregation of misfolded proteins in conditions like AD and PD [188].

Furthermore, calcium and metal ion homeostasis are closely linked to mitochondrial function [189]. Mitochondria rely on finely tuned calcium signaling to regulate ATP production and apoptosis. An imbalance in mitochondrial calcium levels is a hallmark of neurodegenerative diseases, leading to mitochondrial dysfunction and increased ROS production [7]. Compounds in *H. erinaceus*, particularly erinacines and hericenones, have been found to support mitochondrial health by modulating ROS levels and improving energy metabolism [190]. This regulation of calcium homeostasis further reinforces the potential neuroprotective benefits of *H. erinaceus*.

## 5. Nutritional and Therapeutic Applications

Rich in bioactive compounds, the lion’s mane mushroom has been widely recognized for its potential benefits in supporting cognitive function, gut health, immune regulation, and antimicrobial activity [58,191,192,193]. As scientific research continues to unveil its vast therapeutic properties, *H. erinaceus* is gaining momentum in the fields of nutraceuticals, functional foods, and natural medicine [194,195].

Due to its exceptional nutritional profile and medicinal properties, *H. erinaceus* has been increasingly incorporated into dietary supplements and functional foods [146]. This mushroom is available in several forms, including capsules and tablets, powdered form, liquid extracts, functional beverages, and protein bars [60,191,193,196,197] (Figure 4).

One of the main reasons for its growing popularity in functional foods is its ability to blend well with several food matrices while retaining its health-promoting properties [198]. Its mild umami flavor allows for easy incorporation into diverse recipes, ranging from soups and stews to protein powders and health drinks [199]. Additionally, *H. erinaceus* is a rich source of essential nutrients [200]. Its high nutrient density, coupled with the presence of bioactive compounds, makes it an ideal candidate for fortifying diets and promoting overall well-being [42,200].

Among the most well-documented benefits of *H. erinaceus* is its potential to support brain health and cognitive function [83,90]. Studies have suggested that supplementation with *H. erinaceus* may improve memory, focus, and learning capacity [86]; protect against neurodegenerative disorders such as AD and PD [84]; enhance nerve regeneration; and support recovery from brain or spinal cord injuries [201]. Additionally, clinical trials have demonstrated its potential in reducing symptoms of mild cognitive impairment (MCI) [131], and alleviating anxiety and depression [200], likely due to its neurotrophic and anti-inflammatory effects [131,202].

The gut microbiome plays a fundamental role in overall health, and *H. erinaceus* has been recognized for its beneficial effects on digestive function [203]. Its prebiotic polysaccharides, particularly β-glucans, serve as substrates for beneficial gut bacteria, promoting microbiota balance and gut integrity [204]. Several studies suggest that *H. erinaceus* helps in the prevention and treatment of gastric ulcers by protecting the mucosal lining and promoting tissue repair [205]; supports intestinal barrier function, reducing gut inflammation and conditions like leaky gut syndrome [132]; exhibits potential against inflammatory bowel diseases (IBD), such as Crohn’s disease and ulcerative colitis, due to its anti-inflammatory and immunomodulatory properties [110]. Moreover, its antimicrobial compounds may help regulate harmful bacteria, such as *H. pylori*, which is associated with gastric ulcers and stomach cancer [151].

## 6. Future Perspectives and Challenges

Currently, *H. erinaceus* is generally recognized as safe (GRAS) when consumed as a food and primarily marketed as a dietary supplement rather than a pharmaceutical drug [110]. Rodent studies indicate that oral administration of *H. erinaceus* does not cause significant organ damage or alter hematological parameters [206]. However, long-term human studies are needed to confirm these findings and establish safe dosage guidelines. However, as with any fungi, those with known allergies to mushrooms should avoid *H. erinaceus* to prevent allergic reactions [110].

To maximize its therapeutic potential, advancements in biotechnology and extraction techniques are essential for enhancing the yield, purity, and bioavailability of its bioactive compounds [207]. Biotechnological innovations, including solid-state and submerged fermentation, offer promising methods for producing large quantities of mushroom biomass and bioactive compounds. These methods allow for controlled cultivation conditions, improved extraction efficiency, and higher concentrations of target metabolites, such as erinacines and hericenones [208,209]. Furthermore, genetic and metabolic engineering techniques could enhance the biosynthesis of key bioactive molecules. By manipulating specific biosynthetic pathways, researchers can increase the yield of desired compounds, optimize their pharmacological activity, and create tailor-made mushroom extracts for targeted health applications [210,211].

Conventional methods, such as hot water and ethanol extraction, often yield crude extracts with variable potency [23,24]. Advanced techniques, including supercritical fluid extraction, ultrasound-assisted extraction, microwave-assisted extraction, and enzyme-assisted extraction, have shown greater efficiency in isolating polysaccharides, terpenoids, and phenolic compounds while preserving their bioactivity [212]. Additionally, the encapsulation of bioactive compounds into nanoparticles or nanoliposomes may enhance their bioavailability, stability, and therapeutic efficacy [213,214].

One of the primary challenges in translating *H. erinaceus* from laboratory research to clinical use is the significant variability in its bioactive compound content [215]. The potency of its extracts can be influenced by multiple factors, including the strain of the mushroom, cultivation conditions (e.g., substrate composition, temperature, humidity), extraction methods, and post-harvest processing. This variability poses a substantial hurdle in ensuring consistent therapeutic effects across different studies and commercial products [216].

Among the alternatives to overcome this issue are the implementation of analytical techniques such as high-performance liquid chromatography (HPLC), mass spectrometry (MS), and nuclear magnetic resonance (NMR), which can help quantify key bioactive components. By establishing minimum effective concentrations of erinacines, hericenones, and polysaccharides, manufacturers can ensure batch-to-batch consistency [217,218]. Additionally, the use of controlled environmental conditions, genetically characterized strains, and defined growth substrates can help produce mushrooms with more uniform bioactive profiles [42,219].

Another alternative are the agencies such as the U. S. Food and Drug Administration (FDA) and European Medicines Agency (EMA) may require clinical validation and quality control measures before approving mushroom-derived formulations for therapeutic use. Future research should focus on optimizing extraction protocols, defining pharmacokinetic properties, and conducting large-scale clinical trials to validate the therapeutic benefits of standardized formulations marketed as a dietary supplement rather than a pharmaceutical drug.

However, despite its long history of traditional use, the regulatory landscape for medicinal mushrooms remains complex and fragmented across different regions [21]. In many countries, *H. erinaceus* is marketed as a dietary supplement rather than a pharmaceutical product, often requiring less stringent safety and efficacy data. In the EU (European Union), medicinal mushrooms are typically categorized under novel foods or food supplements, depending on the product composition and claims made (Regulation (EU) 2015/2283) [220]. Health claims must comply with the European Food Safety Authority (EFSA) regulations, which require scientific substantiation before approval. However, no specific health claims for *H. erinaceus* have been authorized to date.

In the United States, the U.S. Food and Drug Administration (FDA) regulates mushroom-based products primarily as dietary supplements under the Dietary Supplement Health and Education Act (DSHEA). This classification permits manufacturers to market *H. erinaceus* products without prior FDA approval, provided they do not make disease-related claims. However, companies must ensure product safety and label accuracy [21].

Concerning China and Japan, both countries have long-standing traditions of using medicinal mushrooms in traditional medicine systems [221]. *H. erinaceus* is included in the Chinese Pharmacopoeia and widely used in traditional Chinese medicine (TCM) [222]. It is commonly incorporated into health foods and Kampo medicine formulations in Japan, where regulatory frameworks are more accommodating toward medicinal mushrooms [223].

## 7. Conclusions

Despite the growing body of evidence supporting the health benefits of *H. erinaceus*, several critical research gaps remain. While preclinical and in vitro studies have demonstrated its neuroprotective, antimicrobial, and immunomodulatory properties, large-scale, well-controlled clinical trials are essential to validate these effects in human populations. Future research should focus on defining optimal dosages, long-term safety, and potential interactions with pharmaceuticals to facilitate its integration into evidence-based medicine. A key challenge in translating *H. erinaceus* into clinical and commercial applications is the lack of standardization in extraction methods and bioactive compound quantification. The variability in its polysaccharide, terpenoid, and phenolic content across different cultivation and processing techniques limits its reproducibility and therapeutic consistency. Establishing standardized protocols for cultivation, extraction, and formulation will be critical to ensuring batch-to-batch consistency and regulatory compliance.

Additionally, *H. erinaceus*’ potential as an adjunct therapy for neurodegenerative diseases, gut health, and antimicrobial resistance requires further investigation. The development of novel delivery systems, such as nanoparticles and encapsulated extracts, may enhance bioavailability and therapeutic efficacy, addressing a significant limitation in its pharmacological application. Moreover, its synergistic effects with conventional antibiotics present a promising strategy to combat antimicrobial resistance, necessitating further studies on its mechanisms of action and clinical relevance. To fully harness the therapeutic potential of *H. erinaceus*, collaborative efforts between researchers, clinicians, and regulatory agencies are needed to drive clinical validation and establish standardized guidelines. By addressing these challenges, *H. erinaceus* could emerge as a scientifically validated functional food and therapeutic agent, contributing to the advancement of natural product-based healthcare solutions.

## Figures and Tables

**Figure 1 nutrients-17-01307-f001:**
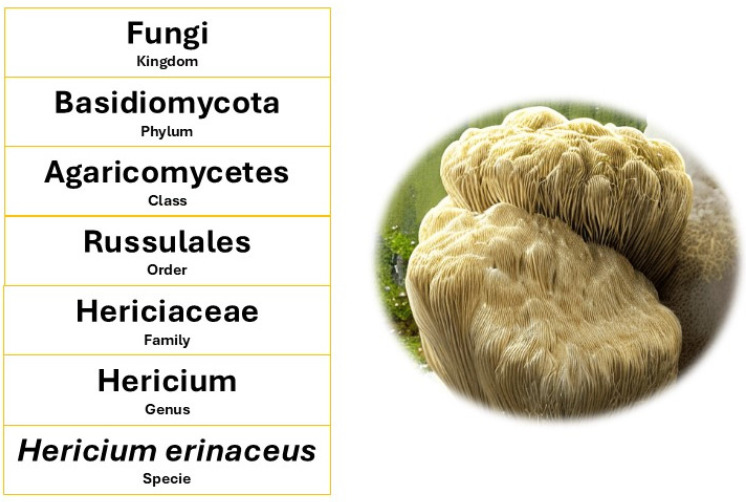
Scientific classification of lion’s mane mushroom (*Hericium erinaceus*). Image created by the authors.

**Figure 2 nutrients-17-01307-f002:**
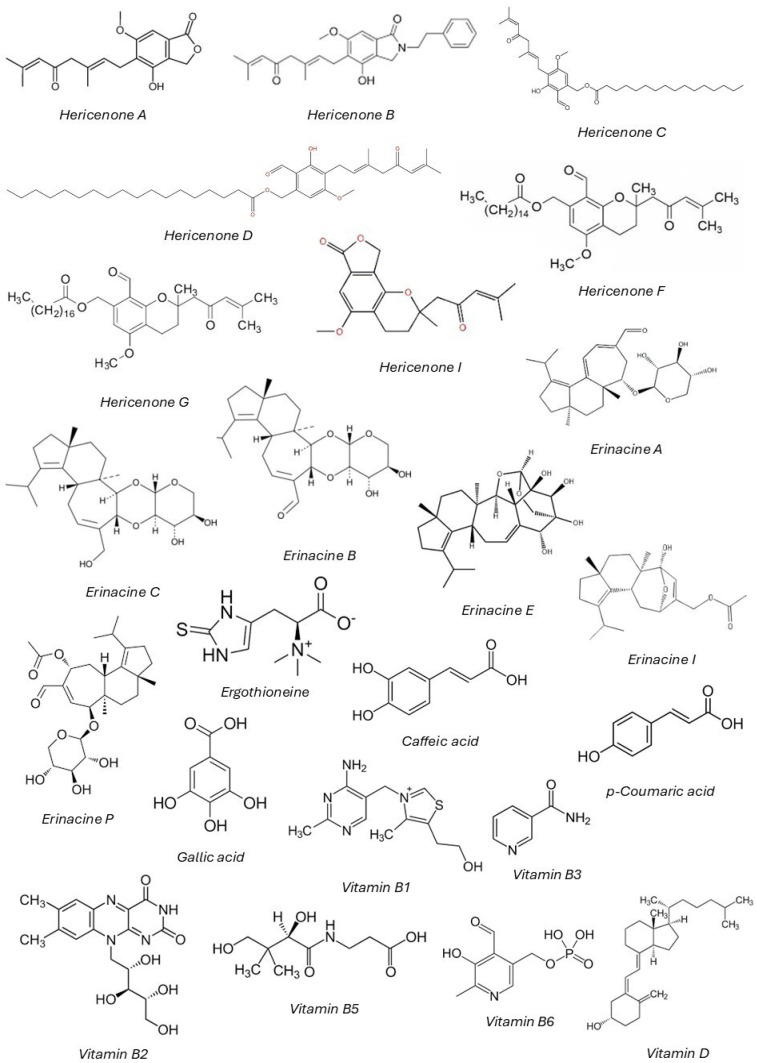
Chemical structures of key bioactive compounds identified in *Hericium erinaceus*, including hericenones, erinacines, and other relevant molecules. These compounds are associated with the mushroom’s neuroprotective, antioxidant, and anti-inflammatory properties.

**Figure 3 nutrients-17-01307-f003:**
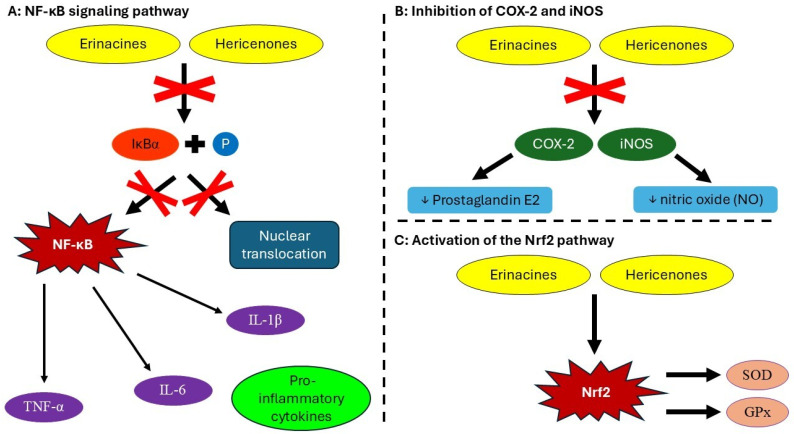
Modulation of signaling pathways and inflammatory mediators by erinacines and hericenones produced by *Hericium erinaceum*. (**A**) NF-κB signaling pathway, where bioactive compounds produced by *H. erinaceus* inhibit the phosphorylation of IκBα, preventing NF-κB activation, which leads to the increased production of cytokines such as TNF-α, IL-6, and IL-1β, and nuclear translocation. (**B**) *H. erinaceus* also acts in the inhibition of COX-2 (cyclooxygenase-2) and iNOS (inducible nitric oxide synthase), where hericenones inhibit COX-2, reducing prostaglandin E2 (PGE2) synthesis, and suppress iNOS expression, leading to reduced nitric oxide (NO) production. (**C**) *H. erinaceus* contributes to anti-inflammatory effects by activating the Nrf2 (nuclear factor erythroid 2–related factor 2) pathway, which enhances the expression of antioxidant enzymes like superoxide dismutase (SOD) and glutathione peroxidase (GPx).

**Figure 4 nutrients-17-01307-f004:**
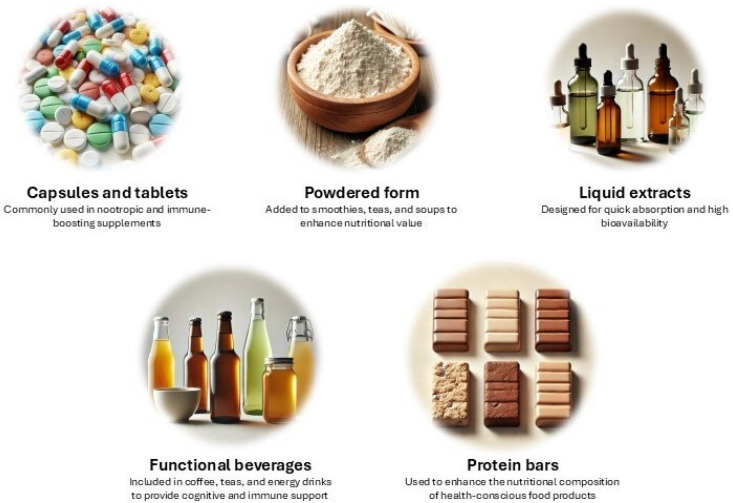
Forms in which *Hericium erinaceus* has been incorporated into dietary supplements and functional foods [60,191,193,196,197].

**Table 1 nutrients-17-01307-t001:** Comparison of *Hericium erinaceus* cultivation methods.

Cultivation Method	Growth Time	Yield	Cost	Difficulty Level	Characteristics	Substrate	Inoculation	Harvest	Application	References
*Log cultivation*	6–12 months	Low	Low	Moderate	Mimics natural habitat; slow but high-quality mushrooms	Hardwood logs (oak, beech, maple) aged for a few weeks	Plug spawn inserted into drilled holes and sealed with wax	Annual harvest for up to 5 years	Traditional, gourmet markets	[52,53,54]
*Sawdust Blocks (Indoor)*	6–8 weeks	High	Medium	High	Fast, controlled conditions, high predictability	Sterilized sawdust with wheat/rice bran and gypsum	Grain spawn mixed into the substrate and incubated for 2–4 weeks	Multiple harvests over a few weeks	Commercial mushroom production	[37,42,51]
*Liquid fermentation*	5–10 days	Very high	High	Advanced	Produces high mycelial biomass and bioactive compounds	Liquid nutrient medium (glucose, yeast extract, peptone)	Inoculated with mycelium and incubated under controlled aeration	Mycelium harvested, dried, and processed	Pharmaceutical, nutraceutical industries	[55,56,57]

**Table 2 nutrients-17-01307-t002:** Main hericenones and erinacines found in *Hericium erinaceus*.

Terpenoid	Chemical Class	Source	Biological Activity	Potential Applications	References
** *Hericenones* **					
*Hericenone A*	Phenolic terpenoid	Fruiting body	Stimulates NGF synthesis, neuroprotective, anti-inflammatory	Neurodegenerative disease prevention, cognitive enhancement	[34,75]
*Hericenone B*	Phenolic terpenoid	Fruiting body	Promotes NGF synthesis, enhances cognitive function, memory improvement	Alzheimer’s treatment, cognitive health	[75,76]
*Hericenone C*	Phenolic terpenoid	Fruiting body	NGF synthesis promotion, neuroprotective, anti-inflammatory	Cognitive disorders, memory loss	[75,77]
*Hericenone D*	Phenolic terpenoid	Fruiting body	Enhances NGF production, antioxidant, neuroprotective	Neurodegeneration prevention, oxidative stress reduction	[75,78]
*Hericenone E*	Phenolic terpenoid	Fruiting body	Neurogenic effects, promotes NGF synthesis	Cognitive decline treatment, neurogenesis	[72,75]
*Hericenone F*	Phenolic terpenoid	Fruiting body	Stimulates NGF synthesis, neuroprotective, enhances brain function	Neuroprotective drugs, cognitive health	[75,79]
*Hericenone G*	Phenolic terpenoid	Fruiting body	Neuroprotective, cognitive enhancement	Nootropic supplements, memory improvement	[80,81]
*Hericenone H*	Phenolic terpenoid	Fruiting body	Anti-inflammatory, promotes nerve regeneration	Nerve damage repair	[34,81]
*Hericenone I*	Phenolic terpenoid	Fruiting body	No protective effect on estrogen receptor stress-dependent cell death	Cardiovascular health, anti-aging formulations	[29]
*Hericenone J*	Phenolic terpenoid	Fruiting body	Enhances NGF expression, neuroprotection	Neurodegenerative disease therapy, brain health	[72,75]
*Hericenone L*	Phenolic terpenoid	Fruiting body	Modulates inflammatory pathways, antioxidant	Chronic inflammation management, metabolic disease therapy	[73,82]
** *Erinacines* **					
*Erinacine A*	Sesquiterpenoid	Mycelium	Potent stimulator of NGF synthesis, enhances neurogenesis, promotes neuronal growth	Alzheimer’s, Parkinson’s disease, cognitive decline	[78,83,84,85]
*Erinacine B*	Sesquiterpenoid	Mycelium	Neuroprotective, enhances NGF synthesis, supports brain health	Cognitive function improvement, neurodegenerative disease	[72,86]
*Erinacine C*	Sesquiterpenoid	Mycelium	Stimulates NGF production, improves cognitive abilities, neurogenesis	Alzheimer’s prevention, brain health	[78,82,87]
*Erinacine D*	Sesquiterpenoid	Mycelium	NGF stimulation, neuroprotective, anti-inflammatory	Neurodegeneration, brain function recovery	[75,78,88]
*Erinacine E*	Sesquiterpenoid	Mycelium	Stimulates NGF synthesis, reduces neuroinflammation	Neuroprotective therapies, cognitive health	[75,89]
*Erinacine F*	Sesquiterpenoid	Mycelium	Promotes NGF production, neurogenesis stimulation	Cognitive disorders, neurodegenerative disease prevention	[75,90]
*Erinacine G*	Sesquiterpenoid	Mycelium	Stimulates NGF synthesis, neuroprotective effects	Neurodegenerative disease prevention, cognitive enhancement	[75,90]
*Erinacine H*	Sesquiterpenoid	Mycelium	Neuroprotective	Brain function recovery	[90]
*Erinacine I*	Sesquiterpenoid	Mycelium	Enhances cognitive function	Memory improvement, Alzheimer’s therapy	[91]
*Erinacine K*	Sesquiterpenoid	Mycelium	Promotes NGF production	Brain health	[75]
*Erinacine P*	Sesquiterpenoid	Mycelium	Potential antimicrobial and anti-inflammatory activity	Antimicrobial applications, immune modulation	[92,93]
*Erinacine Q*	Sesquiterpenoid	Mycelium	Neurotrophic effects, supports neuronal health	Nerve growth support, neurodegeneration prevention	[72,94]
*Erinacine R*	Sesquiterpenoid	Mycelium	Cognitive enhancement	Alzheimer’s treatment	[95]
*Erinacine S*	Sesquiterpenoid	Mycelium	Neuroprotective, improves memory function	Memory preservation, learning enhancement	[72,96]
*Erinacine V*	Sesquiterpenoid	Mycelium	Antioxidant, potential cognitive enhancer	Antioxidant therapy, cognitive support	[73,97]
*Erinacine Z1*	Sesquiterpenoid	Mycelium	Increase the expression of this neurotrophin, regulating inflammatory processes	Inflammation control, neuroprotective drug development	[72]
*Erinacine Z2*	Sesquiterpenoid	Mycelium	Potential application in neurodegenerative disease therapy	Potential Alzheimer’s and Parkinson’s therapy	[93,98]

## Data Availability

This narrative review is based on a comprehensive analysis of previously published studies and does not involve original data collection.

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
