# Peer review of "Lion’s Mane Mushroom (Hericium erinaceus): A Neuroprotective Fungus with Antioxidant, Anti-Inflammatory, and Antimicrobial Potential—A Narrative Review"

_nutrients, 2025, doi:10.3390/nu17081307_

Round 1

Reviewer 1 Report

Comments and Suggestions for Authors

The manuscript presents a well-organized and comprehensive review of Hericium erinaceus (Lion’s Mane mushroom), focusing on its bioactive compounds, neuroprotective potential, antioxidant activity, and antimicrobial effects. The discussion integrates nutritional, biochemical, and pharmacological insights, making it valuable for researchers in the fields of natural products, neuroprotection, and functional foods. The review is well-referenced with a broad range of recent studies, indicating a strong literature foundation.

The authors successfully highlight potential therapeutic applications and discuss mechanistic pathways, such as nerve growth factor (NGF) stimulation, oxidative stress modulation, and immune response regulation. Additionally, the inclusion of cultivation techniques and regulatory challenges adds a practical dimension to the paper.

However, several areas require improvement to enhance readability and scientific rigor.

Introduction

  • The introduction could briefly mention major bioactive compounds (e.g., polysaccharides, terpenoids) to give a clearer overview upfront.
  • The discussion of chronic diseases linked to oxidative stress and inflammation is relevant but could benefit from a more concise presentation to maintain focus on H. erinaceus.
  • No clear research question or objective statement is present. A sentence explicitly stating the main goals of the review would improve clarity.

Chemical Composition and Bioactive Compounds

  • Too much technical detail without synthesis: The section lists multiple chemical constituents (e.g., hericenones, erinacines) without discussing their relative importance in different biological effects.
  • A comparison with other medicinal mushrooms (e.g., Ganoderma lucidum, Cordyceps sinensis) would enhance the reader’s understanding of H. erinaceus’s unique bioactivity profile.

Neuroprotective Effects

  • Mechanisms need clearer linking to clinical relevance: While the review discusses NGF stimulation and neuroinflammation reduction, it lacks direct connections to human diseases like Alzheimer’s or Parkinson’s.
  • More emphasis on human studies is needed: Most findings are based on animal or in vitro studies. Clinical trials should be summarized separately to highlight their significance.
  • A discussion on bioavailability and blood-brain barrier penetration would be useful, as this is a key challenge in translating neuroprotective compounds into therapies.

Antimicrobial Properties

  • More specific mechanisms of action should be included (e.g., cell membrane disruption, inhibition of biofilm formation).
  • Comparisons with conventional antibiotics are missing: Does H. erinaceus exhibit bactericidal or bacteriostatic effects? How does its activity compare to known antimicrobials?
  • Clinical applications are unclear: While antimicrobial potential is emphasized, how feasible is it to use H. erinaceus in drug formulations or food preservation?

Regulatory Challenges and Future Perspectives

  • Standardization Issues: Given that mushroom extracts vary significantly in potency, how can standardization improve clinical efficacy?
  • Safety and Toxicity Considerations: Are there any reports of adverse effects or contraindications?
  • Commercialization Potential: Is H. erinaceus currently marketed as a pharmaceutical in any country?

Conclusion

  • Instead of repeating previous points, it should emphasize research gaps and future directions.
  • A call for clinical trials and standardization efforts would strengthen the impact.

While the manuscript is scientifically rich and well-researched, it requires better structuring, improved critical analysis, and deeper discussions on clinical applications. Expanding human trial data, reducing redundancy, and addressing standardization issues will significantly improve its impact.

Author Response

Response to reviewers’ comments:

Reviewer 01:

The manuscript presents a well-organized and comprehensive review of Hericium erinaceus (Lion’s Mane mushroom), focusing on its bioactive compounds, neuroprotective potential, antioxidant activity, and antimicrobial effects. The discussion integrates nutritional, biochemical, and pharmacological insights, making it valuable for researchers in the fields of natural products, neuroprotection, and functional foods. The review is well-referenced with a broad range of recent studies, indicating a strong literature foundation.

The authors successfully highlight potential therapeutic applications and discuss mechanistic pathways, such as nerve growth factor (NGF) stimulation, oxidative stress modulation, and immune response regulation. Additionally, the inclusion of cultivation techniques and regulatory challenges adds a practical dimension to the paper.

R.: Thank you for your thoughtful and positive feedback on our manuscript. We truly appreciate your recognition of the comprehensive nature of our review. Your insights reinforce our commitment to delivering a well-structured and informative review, and we are grateful for the opportunity to contribute to the scientific community. Thank you once again for your time and valuable evaluation.

However, several areas require improvement to enhance readability and scientific rigor.

R.: Thank you for your constructive feedback and for highlighting areas for improvement. We greatly appreciate your insights, which have helped us refine the manuscript to enhance both readability and scientific rigor. We have carefully considered your suggestions and made the necessary revisions to address them to the best of our ability. We sincerely appreciate your time and effort in reviewing our work and for contributing to its improvement.

Introduction

  • The introduction could briefly mention major bioactive compounds (e.g., polysaccharides, terpenoids) to give a clearer overview upfront.

R.: We appreciate your suggestion and have incorporated a brief mention of the major bioactive compounds, such as polysaccharides and terpenoids, in the introduction to provide a clearer overview upfront.

Lines 108-114: H. erinaceus contains a variety of bioactive compounds, which include polysaccharides, terpenoids, and phenolic compounds. Polysaccharides, particularly β-glucans, are known for their immunomodulatory and neuroprotective effects. Terpenoids, such as hericenones and erinacines, have been shown to stimulate nerve growth factor (NGF) synthesis, promoting neuronal growth and repair. Additionally, phenolic compounds present in H. erinaceus exhibit strong antioxidant properties, helping to mitigate oxidative stress and inflammation [29].

  • The discussion of chronic diseases linked to oxidative stress and inflammation is relevant but could benefit from a more concise presentation to maintain focus on H. erinaceus.

R.: Thank you for your suggestion. We have revised the discussion on chronic diseases linked to oxidative stress and inflammation to make it more concise and maintain the focus on H. erinaceus. Specifically, lines 85-96 have been removed to improve clarity and streamline the presentation.

  • No clear research question or objective statement is present. A sentence explicitly stating the main goals of the review would improve clarity.

R.: Thank you for your suggestion. We have added a clear research question and objective statement to explicitly define the main goals of the review, improving clarity and direction.

Lines 126-129: Specifically, this narrative review seeks to provide a comprehensive understanding of the mechanisms through which H. erinaceus exerts its effects, summarize current scientific findings, and identify potential gaps in knowledge that warrant further research.

Chemical Composition and Bioactive Compounds

  • Too much technical detail without synthesis: The section lists multiple chemical constituents (e.g., hericenones, erinacines) without discussing their relative importance in different biological effects.

R.: Thank you for your valuable feedback. We have revised the section to provide a better synthesis of the chemical constituents, emphasizing their relative importance in different biological effects rather than merely listing them (Lines 202-292).

  • A comparison with other medicinal mushrooms (e.g., Ganoderma lucidumCordyceps sinensis) would enhance the reader’s understanding of H. erinaceus’s unique bioactivity profile.

R.: We appreciate your suggestion. A comparison was included not only with Ganoderma lucidum and Cordyceps sinensis, as suggested, but also with other medicinal mushrooms such as Lentinula edodes (shiitake) and Phellinus linteus, to provide a more comprehensive understanding of H. erinaceus’s unique bioactivity profile.

Lines 279-292: Compared to other well-known medicinal mushrooms, H. erinaceus stands out primarily for its neuroprotective properties, largely attributed to its unique erinacines and hericenones [72, 78, 93]. Ganoderma lucidum (reishi), for instance, is widely recognized for its strong immunomodulatory and anti-cancer effects due to its high content of triterpenoids and polysaccharides [103]. Cordyceps sinensis (now Ophiocordyceps sinensis), another medicinal mushroom, is known for its energy-boosting and anti-fatigue properties, largely attributed to cordycepin and adenosine derivatives, which enhance mitochondrial function [104]. Meanwhile, Lentinula edodes (shiitake) is particularly rich in lentinan, a β-glucan with immunomodulatory and anti-cancer properties [105]. Another notable species, Phellinus linteus, exhibits potent anti-tumor and anti-inflammatory activities through its unique polyphenolic compounds [22]. Unlike these mushrooms, H. erinaceus remains unparalleled in its ability to stimulate NGF synthesis, making it a promising candidate for neurodegenerative disease treatment and cognitive enhancement [72, 78]. This distinct biochemical profile underscores its unique therapeutic niche among medicinal fungi.

Neuroprotective Effects

  • Mechanisms need clearer linking to clinical relevance: While the review discusses NGF stimulation and neuroinflammation reduction, it lacks direct connections to human diseases like Alzheimer’s or Parkinson’s.

R.: We appreciate this suggestion. The discussion on NGF stimulation and neuroinflammation reduction has been revised to establish clearer links to clinical relevance, specifically addressing their implications for human diseases such as Alzheimer’s and Parkinson’s (Lines 305-335)

  • More emphasis on human studies is needed: Most findings are based on animal or in vitro studies. Clinical trials should be summarized separately to highlight their significance.

R.: We appreciate this suggestion. More emphasis has been placed on human studies, and a new section titled "Clinical Trials" (Lines 361-402) has been added to separately summarize and highlight the significance of available clinical research.

  • A discussion on bioavailability and blood-brain barrier penetration would be useful, as this is a key challenge in translating neuroprotective compounds into therapies.

R.: Thank you for this valuable suggestion. A new section titled "Bioavailability and Blood-Brain Barrier Penetration" (Lines 403-425) has been added to address this crucial challenge in translating neuroprotective compounds into therapies.

Antimicrobial Properties

  • More specific mechanisms of action should be included (e.g., cell membrane disruption, inhibition of biofilm formation).

R.: Thank you for your valuable suggestion. We have expanded the discussion on the mechanisms of action, including more specific details such as cell membrane disruption and inhibition of biofilm formation, as requested.

Lines 470-500:

  1. Cell Membrane Disruption: several bioactive compounds in erinaceus, particularly terpenoids and phenolic compounds, have been shown to interfere with bacterial and fungal cell membranes [29, 74]. These compounds disrupt membrane integrity by altering lipid bilayer stability, leading to increased permeability, leakage of intracellular contents, and eventual cell death. This mechanism is particularly relevant against Gram-positive bacteria, which have a thick peptidoglycan layer that is more susceptible to membrane-targeting agents [154].
  2. Inhibition of Biofilm Formation: biofilms are protective structures formed by microbial communities that enhance resistance to antibiotics and immune responses [155]. Polysaccharides and terpenoids from erinaceus have demonstrated the ability to inhibit biofilm formation by interfering with quorum sensing pathways, the bacterial communication system that regulates biofilm development [93]. By preventing biofilm maturation, H. erinaceus compounds enhance the susceptibility of bacteria to antimicrobial agents and host immune defenses [155].
  3. Enzyme Inhibition and Metabolic Disruption: phenolic compounds in erinaceus have been reported to inhibit key bacterial enzymes involved in cell wall synthesis, DNA replication, and energy metabolism [69, 156]. For example, some erinacines and hericenones have been shown to interfere with bacterial ATP production, disrupting essential metabolic pathways and leading to growth inhibition [93, 145].
  4. Induction of Oxidative Stress: some bioactive compounds in erinaceus promote the generation of reactive oxygen species in microbial cells [157]. Excess ROS accumulation leads to oxidative damage to proteins, lipids, and DNA, ultimately resulting in cell death [7]. This mechanism is particularly effective against antibiotic-resistant bacteria, which often rely on antioxidant defense systems to survive in hostile environments [142].
  5. Modulation of Host Immune Responses: polysaccharides, especially β-glucans, play a crucial role in enhancing the host's immune response against infections [67]. These compounds stimulate macrophages, dendritic cells, and NK cells, boosting antimicrobial activity and facilitating the clearance of bacterial and fungal pathogens [110].
  • Comparisons with conventional antibiotics are missing: Does H. erinaceus exhibit bactericidal or bacteriostatic effects? How does its activity compare to known antimicrobials?

R.: Thank you for your insightful comment. We have addressed this question in lines 507-492, where we discuss whether H. erinaceus exhibits bactericidal or bacteriostatic effects and compare its activity to conventional antibiotics, as shown below.

Lines 475-424: The antimicrobial activity of H. erinaceus varies depending on the specific bioactive compounds and target microorganisms. In general, its effects are predominantly bacteriostatic, meaning it inhibits bacterial growth rather than directly killing bacteria [164]. This contrasts with many conventional antibiotics, such as β-lactams (e.g., penicillins and cephalosporins), which exhibit bactericidal activity by disrupting bacterial cell walls and causing lysis [165]. However, some studies have reported bactericidal effects of H. erinaceus extracts, particularly against Gram-positive bacteria like S. aureus [166] and B. subtilis [167]. These effects are likely due to membrane-disrupting terpenoids, which resemble the mechanism of antimicrobial peptides [159] and some lipopeptide antibiotics (e.g., daptomycin) [168]. The ability of H. erinaceus to disrupt bacterial membranes suggests a mode of action similar to that of polymyxins, which are effective against Gram-negative bacteria, but with a different structural basis [168].

When compared to well-known antibiotics, H. erinaceus polysaccharides function similarly to macrolides (e.g., erythromycin) by inhibiting bacterial protein synthesis and biofilm formation, leading to reduced bacterial virulence [169]. Phenolic compounds in H. erinaceus exhibit antioxidant and antimicrobial properties comparable to quinolones (e.g., ciprofloxacin), which target bacterial DNA replication [170]. While terpenoids from H. erinaceus act similarly to lipopeptide antibiotics by disrupting bacterial membranes [171].

  • Clinical applications are unclear: While antimicrobial potential is emphasized, how feasible is it to use H. erinaceus in drug formulations or food preservation?

R.: Thank you for your valuable comment. We have clarified the feasibility of using H. erinaceus in drug formulations and food preservation in the revised text. This information has been added and can be found in lines 531-546, as shown below.

Lines 531-546: While H. erinaceus has shown promise as an antimicrobial agent, challenges such as bioavailability and extraction efficiency need to be addressed before it can be incorporated into mainstream pharmaceuticals [177]. Given its broad antimicrobial spectrum and potential to enhance antibiotic efficacy, H. erinaceus holds promise for several applications, such as its use as an adjuvant therapy to enhance conventional antibiotics and potential application in topical antimicrobial agents for wound healing and skin infections [175]. Additionally, H. erinaceus extracts also could be used as natural preservatives to extend shelf life and prevent microbial contamination in food products [162]. Studies have shown that mushroom-derived bioactives can inhibit the growth of foodborne pathogens such as Listeria monocytogenes and Salmonella spp. without the need for synthetic preservatives [176]. However, optimization of extraction methods and stability studies under different storage conditions are necessary to ensure practical implementation [177]. Despite these promising applications, further research is needed to determine the most effective delivery methods, safety profiles, and regulatory pathways for integrating H. erinaceus into clinical and industrial products [178]. Large-scale clinical trials and toxicological studies will be essential to validate its use as a therapeutic or preservative agent [142].

Regulatory Challenges and Future Perspectives

  • Standardization Issues: Given that mushroom extracts vary significantly in potency, how can standardization improve clinical efficacy?

R.: Thank you for your insightful comment. We have addressed the issue of standardization and its role in improving clinical efficacy in the revised text. This discussion has been added and can be found in lines 652-673, as shown below.

Lines 652-673: One of the primary challenges in translating H. erinaceus from laboratory research to clinical use is the significant variability in its bioactive compound content [215]. The potency of its extracts can be influenced by multiple factors, including the strain of the mushroom, cultivation conditions (e.g., substrate composition, temperature, humidity), extraction methods, and post-harvest processing. This variability poses a substantial hurdle in ensuring consistent therapeutic effects across different studies and commercial products [216].

 Among the alternatives to overcome this issue are the implementation of analytical techniques such as high-performance liquid chromatography (HPLC), mass spectrometry (MS), and nuclear magnetic resonance (NMR) can help quantify key bioactive components. By establishing minimum effective concentrations of erinacines, hericenones, and polysaccharides, manufacturers can ensure batch-to-batch consistency [217, 218]. Additionally, the use of controlled environmental conditions, genetically characterized strains, and defined growth substrates can help produce mushrooms with more uniform bioactive profiles [42, 219].

Another alternative are the agencies such as the U. S. Food and Drug Administration (FDA) and European Medicines Agency (EMA) may require clinical validation and quality control measures before approving mushroom-derived formulations for therapeutic use. Future research should focus on optimizing extraction protocols, defining pharmacokinetic properties, and conducting large-scale clinical trials to validate the therapeutic benefits of standardized formulations to marketed as a dietary supplement rather than a pharmaceutical drug.

  • Safety and Toxicity Considerations: Are there any reports of adverse effects or contraindications?

R.: Thank you for your important observation. We have included a discussion on safety and toxicity considerations, including reports of adverse effects and contraindications. This information can be found in lines 627-633, as shown below.

Lines 627-633: Currently, H. erinaceus is generally recognized as safe (GRAS) when consumed as food and primarily marketed as a dietary supplement rather than a pharmaceutical drug [110]. Rodent studies indicate that oral administration of H. erinaceus does not cause significant organ damage or alter hematological parameters [206]. However, long-term human studies are needed to confirm these findings and establish safe dosage guidelines. However, as with any fungi, those with known allergies to mushrooms should avoid H. erinaceus to prevent allergic reactions [110].

  • Commercialization Potential: Is H. erinaceus currently marketed as a pharmaceutical in any country?

R.: Thank you for your valuable question. H. erinaceus is widely available in the global market, primarily as a dietary supplement and functional food ingredient rather than a registered pharmaceutical. However, in some countries, particularly in Asia (e.g., China, Japan, and South Korea), it is incorporated into traditional medicine formulations with claims related to cognitive health and neuroprotection. Despite its recognized bioactive properties, regulatory approval as a pharmaceutical remains limited, and more clinical trials are needed to support its medicinal use.

Conclusion

  • Instead of repeating previous points, it should emphasize research gaps and future directions.
  • A call for clinical trials and standardization efforts would strengthen the impact.

R.: Thank you for your valuable feedback. The conclusion has been rewritten to emphasize research gaps and future directions, rather than repeating previous points. Additionally, we have included a call for clinical trials and standardization efforts to strengthen the impact of the discussion.

Lines 720-743: Despite the growing body of evidence supporting the health benefits of H. erinaceus, several critical research gaps remain. While preclinical and in vitro studies have demonstrated its neuroprotective, antimicrobial, and immunomodulatory properties, large-scale, well-controlled clinical trials are essential to validate these effects in human populations. Future research should focus on defining optimal dosages, long-term safety, and potential interactions with pharmaceuticals to facilitate its integration into evidence-based medicine. A key challenge in translating H. erinaceus into clinical and commercial applications is the lack of standardization in extraction methods and bioactive compound quantification. The variability in polysaccharide, terpenoid, and phenolic content across different cultivation and processing techniques limits reproducibility and therapeutic consistency. Establishing standardized protocols for cultivation, extraction, and formulation will be critical to ensuring batch-to-batch consistency and regulatory compliance.

Additionally, H. erinaceus' potential as an adjunct therapy for neurodegenerative diseases, gut health, and antimicrobial resistance requires further investigation. The development of novel delivery systems, such as nanoparticles and encapsulated extracts, may enhance bioavailability and therapeutic efficacy, addressing a significant limitation in its pharmacological application. Moreover, its synergistic effects with conventional antibiotics present a promising strategy to combat antimicrobial resistance, necessitating further studies on its mechanisms of action and clinical relevance. To fully harness the therapeutic potential of H. erinaceus, collaborative efforts between researchers, clinicians, and regulatory agencies are needed to drive clinical validation and establish standardized guidelines. By addressing these challenges, H. erinaceus could emerge as a scientifically validated functional food and therapeutic agent, contributing to the advancement of natural product-based healthcare solutions.

While the manuscript is scientifically rich and well-researched, it requires better structuring, improved critical analysis, and deeper discussions on clinical applications. Expanding human trial data, reducing redundancy, and addressing standardization issues will significantly improve its impact.

R.: Thank you for your thoughtful and constructive feedback. We sincerely appreciate your insights and suggestions, which have helped us refine and strengthen the manuscript. We have made every possible effort to address your recommendations by improving the structure, enhancing critical analysis, and expanding discussions on clinical applications, human trial data, and standardization issues. Your valuable input has been instrumental in enhancing the overall quality and impact of our work.

Reviewer 2 Report

Comments and Suggestions for Authors

After the following revisions, the work conducted by Contato and Conte-Junior can be considered for publication in Nutrients:

Dear authors, you should include the type of review you conducted in the title, in the abstract and the whole manuscript.

The abstract has to be structured. The searched databases and applied methods are missing, as well as the highlighting of the most relevant results and directions to be followed in future researches.

The Introduction is adequate, but the study’s objectives need to be clearly stated.

A Methods section is missing. Please, pay attention to section 2 of this paper: https://www.mdpi.com/2304-8158/10/6/1175

Figure 1: Do you have the copyrights for this figure? Please, mention it. The name of the species has to be in italics.

Sections 3.2 and 3.3 should be expanded and more evidence to document the antioxidant and antimicrobial activities should be provided.

Where can we find this mushroom available worldwide. This perspective should be explored and analyzed in section 4 or by adding a new section with this purpose.

Conclusions are adequate.

Author Response

Response to reviewers’ comments:

Reviewer 02:

After the following revisions, the work conducted by Contato and Conte-Junior can be considered for publication in Nutrients.

R.: The authors appreciate you for your time and review; and hope to promptly respond to the reviewer’s suggestions and considerations in this new round.

Dear authors, you should include the type of review you conducted in the title, in the abstract and the whole manuscript.

R.: Thank you for your comment and suggestion. As recommended, we have included the description of the type of review performed in the manuscript. We now specify that it is a narrative review in the title, abstract, and the whole manuscript.

The abstract has to be structured. The searched databases and applied methods are missing, as well as the highlighting of the most relevant results and directions to be followed in future researches.

R.: Thank you for your valuable feedback. The abstract has been rewritten to meet the requirements. We have structured it accordingly, included the searched databases and applied methods, and highlighted the most relevant results as well as future research directions.

Lines 22-44: Hericium erinaceus, commonly known as lion’s mane mushroom, has gained increasing scientific interest due to its rich composition of bioactive compounds and diverse health-promoting properties. This narrative review provides a comprehensive overview of the nutritional and therapeutic potential of H. erinaceus, with a particular focus on its anti-inflammatory, antioxidant, and antimicrobial activities. A structured literature search was performed using databases such as PubMed, Scopus, Science Direct, Web of Science, Science Direct, and Google Scholar. Studies published in the last two decades focusing on H. erinaceus' bioactive compounds were included. Studies published in the last decade focusing on H. erinaceus' bioactive compounds were included. The chemical composition of H. erinaceus includes polysaccharides, terpenoids (hericenones and erinacines), and phenolic compounds, which exhibit potent antioxidant effects by scavenging reactive oxygen species (ROS) and inducing endogenous antioxidant enzymes. Additionally, H. erinaceus shows promising antimicrobial activity against bacterial and fungal pathogens, with potential applications in combating antibiotic-resistant infections. The mushroom's capacity to stimulate nerve growth factor (NGF) synthesis has highlighted its potential in preventing and managing neurodegenerative diseases, such as Alzheimer's and Parkinson's. Advances in biotechnological methods, including optimized cultivation techniques and novel extraction methods, may further enhance the bioavailability and pharmacological effects of H. erinaceus. Despite promising findings, clinical validation remains limited. Future research should prioritize large-scale clinical trials, standardization of extraction methods, and elucidation of pharmacokinetics to facilitate its integration into evidence-based medicine. The potential of H. erinaceus as a functional food, nutraceutical, and adjunct therapeutic agent highlights the need for interdisciplinary collaboration between researchers, clinicians, and regulatory bodies.

The Introduction is adequate, but the study’s objectives need to be clearly stated.

R.: Thank you for your suggestion. We have added a clear research question and objective statement to explicitly define the main goals of the review, improving clarity and direction.

Lines 126-129: Specifically, this narrative review seeks to provide a comprehensive understanding of the mechanisms through which H. erinaceus exerts its effects, summarize current scientific findings, and identify potential gaps in knowledge that warrant further research.

A Methods section is missing. Please, pay attention to section 2 of this paper: https://www.mdpi.com/2304-8158/10/6/1175

R.: Thank you for your suggestion. A section titled "Methods" has been added to the manuscript, following the structure of the referenced article.

Lines 130-146: This narrative review was performed following three steps: conducting the search, reviewing abstracts and full texts, and discussing the results. For this, the PubMed, Scopus, Science Direct, Web of Science, Science Direct, and Google Scholar databases were searched to identify relevant studies, according to the development of the review. The final search was conducted in March 2025 and included international English-language articles, online reports, and electronic books. The keyword “Hericium erinaceus” was used in combination with other terms such as characteristics, habitat, chemical composition, cultivation methods, anti-inflammatory activity, clinical trials, bioavailability, blood-brain barrier penetration, antioxidant activity, antimicrobial activity, calcium binding activity, nutritional and therapeutic applications, challenges, or regulation. After the complete search, the abstracts were read to ensure that they addressed the topic of interest. All duplicates were removed, and the abstracts of the remaining articles were reviewed to ensure that they addressed the inclusion criteria of the review. The eligible criteria were studies that analyzed Hericium erinaceus in combination with the other terms mentioned above. Therefore, the studies of interest were summarized and synthesized to integrate the narrative review. Since it is a narrative review, it was not necessary to document the literature search in specific platforms [36]

Figure 1: Do you have the copyrights for this figure? Please, mention it. The name of the species has to be in italics.

R.: Thank you for your valuable feedback. We confirm that the figure was created by the authors, and we have now formatted the specie name in italics as recommended. We appreciate your careful review and suggestions.

Sections 3.2 and 3.3 should be expanded and more evidence to document the antioxidant and antimicrobial activities should be provided.

R.: Thank you for your insightful suggestion. We have expanded the sections, providing additional evidence to better document the antioxidant and antimicrobial activities. Your detailed feedback has been invaluable in strengthening the manuscript, and we truly appreciate your time and expertise in reviewing our work.

Where can we find this mushroom available worldwide. This perspective should be explored and analyzed in section 4 or by adding a new section with this purpose.

R.: Thank you very much for your thoughtful suggestion. We appreciate your keen attention to detail and your effort in helping us improve the manuscript. The information regarding the global availability of Hericium erinaceus was already included in Section 3.2, "Habitat and cultivation methods of H. erinaceus” (Lines 180-183). However, we have now expanded and refined this section to provide greater clarity and emphasis on this important aspect. Additionally, we highlight that this mushroom is already cultivated and consumed worldwide, including in Brazil. We sincerely appreciate your valuable feedback, which has been instrumental in enhancing the quality of our work.

Lines 180-183: This species prefers temperate forests in North America, Europe, and Asia, thriving in regions with high humidity and moderate temperatures, but it has already expanded to the most diverse regions of the planet and is marketed globally [21, 45].

Conclusions are adequate.

R.: Thank you for your thoughtful and constructive feedback. We sincerely appreciate your insights and suggestions, which have helped us refine and strengthen the manuscript. We have made every possible effort to address your recommendations. Your valuable input has been instrumental in enhancing the overall quality and impact of our work.

Reviewer 3 Report

Comments and Suggestions for Authors

The manuscript „ Lion’s mane mushroom (Hericium erinaceus): a neuroprotective  fungus with antioxidant, anti-inflammatory, and antimicrobial  potential” highlights a growing interest in edible mushrooms, extending from their culinary value to health benefits. The numerous health supplements containing powdered mushrooms or mushroom extracts are offered by various companies. Providing reliable information on composition and properties of Lion’s mane mushroom is important for general public and especially scientific community, coordinating research and indicating promising research targets.

In recent years, there were several review papers published, concerning Lion’s mane mushroom. The Authors included some of them in their reference list, as well as mentioned the increased interest in the subject.

The content of the submitted manuscript is arranged in four topics: neuroprotection, antioxidation, anti-inflammation and antimicrobial activity. However, the list of indicated components of Lion’s mane mushroom is limited. Polysaccharides and terpenoids are described, with Table 2 listing hericenones and erinacines. Bioactive proteins are mentioned in the text, without elaboration. Some other interesting compounds, either characteristic for Lion’s mane mushroom, or of general interest (ergothioneine), are not included. There are no structures shown in the text. The concentration of discussed compounds is also not mentioned.

The description of biological activities is condensed, with multiple examples. Some activities seem to be associated with the components not discussed in the paper. Some activities (calcium binding etc.) are not discussed.

The part of the paper, presenting the cultivation methods (including consumer-accessible variant), supplement market and regulations is really valuable, benefiting both users and researchers and industry. This material is unique among the published reviews and greatly increases the importance of the current manuscript.

In general, the text is prepared in an interesting way, with minor issues (minerals like selenium, zinc, and potassium (line 71) – minerals, sources of…?; unfortunate answer for data availability in line 432). Please check the last line in Figure 1 (Specie).

Author Response

Response to reviewers’ comments:

Reviewer 03:

The manuscript “Lion’s mane mushroom (Hericium erinaceus): a neuroprotective  fungus with antioxidant, anti-inflammatory, and antimicrobial  potential” highlights a growing interest in edible mushrooms, extending from their culinary value to health benefits. The numerous health supplements containing powdered mushrooms or mushroom extracts are offered by various companies. Providing reliable information on composition and properties of Lion’s mane mushroom is important for general public and especially scientific community, coordinating research and indicating promising research targets.

In recent years, there were several review papers published, concerning Lion’s mane mushroom. The Authors included some of them in their reference list, as well as mentioned the increased interest in the subject.

R.: R.: The authors appreciate you for your time and review; and hope to promptly respond to the reviewer’s suggestions and considerations in this new round. Thank you for your thoughtful comments and for highlighting the growing interest in Hericium erinaceus and its health benefits. We appreciate your recognition of the importance of providing reliable information on its composition and bioactive properties. Additionally, we acknowledge the existence of previous review papers on this topic and have aimed to build upon them by offering a comprehensive and updated perspective, addressing key gaps and recent advancements in the field. Your feedback is greatly valued.

The content of the submitted manuscript is arranged in four topics: neuroprotection, antioxidation, anti-inflammation and antimicrobial activity. However, the list of indicated components of Lion’s mane mushroom is limited. Polysaccharides and terpenoids are described, with Table 2 listing hericenones and erinacines. Bioactive proteins are mentioned in the text, without elaboration.

R.: Thank you for your valuable feedback. In response, we have expanded our discussion on bioactive proteins and have incorporated this information in lines 207-216. We hope this addition strengthens the manuscript and provides a more comprehensive overview of the key bioactive compounds found in H. erinaceus.

Lines 207-216: Some examples of bioactive proteins found include lectins, carbohydrate-binding proteins which exhibit immunomodulatory and antimicrobial activities [61], as well as glucanases and chitinases, enzymes that degrade fungal polysaccharides and stimulate immune responses [62]. Oxidative enzymes involved in lignin degradation and detoxification like laccases and peroxidases, have demonstrated antioxidant and antibacterial properties [63], while ribosome-inactivating proteins (RIPs) can inhibit protein synthesis in target cells, potentially exhibiting cytotoxic effects against tumors or pathogens [64]. Additionally, hydrophobins are surface-active proteins that play roles in fungal adhesion and biofilm formation, with emerging biomedical applications [65].

Some other interesting compounds, either characteristic for Lion’s mane mushroom, or of general interest (ergothioneine), are not included.

R.: Thank you for your insightful comment. We appreciate your suggestion and have now included a discussion on ergothioneine and other relevant compounds in lines 247-253. We believe this addition enhances the comprehensiveness of our manuscript.

Lines 247-253: Moreover, H. erinaceus contains ergothioneine, a histidine-derived amino acid with potent antioxidant properties [99]. Ergothioneine has garnered increasing interest due to its ability to neutralize ROS and reduce oxidative stress in neuronal cells. Unlike many dietary antioxidants, ergothioneine is actively transported into cells via the OCTN1 transporter, granting it distinct bioavailability [100]. Studies suggest that its neuroprotective action may have implications in preventing neurodegenerative diseases such AD and PD, where oxidative stress plays a central role [78, 99].

There are no structures shown in the text.

R.: Thank you for your valuable feedback. We have addressed this concern by adding a new figure (Figure 2) that presents the chemical structures of key bioactive compounds identified in Hericium erinaceus (Line 275). We believe this inclusion improves the clarity and completeness of our manuscript.

The concentration of discussed compounds is also not mentioned.

R.: Thank you for your insightful comment. We have now included information on the concentrations of the compounds discussed based on the most recent studies available in literature (Lines 254-262). We believe this addition enhances the comprehensiveness of our discussion.

Lines 254-262: The biological activity of H. erinaceus is directly related to the concentration of its active compounds. Studies indicate that the levels of hericenones and erinacines vary depending on the cultivation substrate and the fungal developmental stage [101]. For instance, hericenones extracted from the fruiting body can be present at concentrations ranging from <20 to 500 µg/g of dry weight, whereas erinacines, found in the mycelium, can reach concentrations ~150 µg/g [90]. Additionally, ergothioneine in H. erinaceus has been detected at levels between 0.34 and 1.30 mg/g, depending on cultivation conditions [78, 100]. Accurate quantification of these compounds is essential for understanding their bioavailability and therapeutic efficacy.

The description of biological activities is condensed, with multiple examples. Some activities seem to be associated with the components not discussed in the paper. Some activities (calcium binding etc.) are not discussed.

R.: Thank you very much for your thoughtful and constructive feedback. We truly appreciate your insights, which have helped us refine and strengthen our manuscript. In response to your comments, we have carefully revised and expanded the text to improve the clarity and depth of our discussion. Additionally, we have included a new section titled "Calcium Binding Activity and Other Functional Properties" (Lines 547-580) to specifically address this aspect. We hope these improvements enhance the overall quality of the manuscript and make it more comprehensive.

The part of the paper, presenting the cultivation methods (including consumer-accessible variant), supplement market and regulations is really valuable, benefiting both users and researchers and industry. This material is unique among the published reviews and greatly increases the importance of the current manuscript.

R.: We sincerely appreciate your kind words and thoughtful evaluation of our work. It is truly gratifying to know that our discussion on cultivation methods, consumer-accessible variants, the supplement market, and regulations adds unique value to literature. Our goal was to provide a comprehensive perspective that benefits not only researchers but also industry professionals and consumers. Thank you immensely for your encouraging feedback.

In general, the text is prepared in an interesting way, with minor issues (minerals like selenium, zinc, and potassium (line 71) – minerals, sources of…?;

R.: Thank you for your careful review and valuable feedback. We have revised and completed the text to clarify the mention of minerals, as can be seen in Lines 79-80. We appreciate your insightful comments, which have helped us improve the manuscript.

Unfortunate answer for data availability in line 432.

R.: Thank you for your observation. We have revised the data availability statement, which now reads: "This narrative review is based on a comprehensive analysis of previously published studies and does not involve original data collection." (Lines 755-757).

Please check the last line in Figure 1 (Specie).

R.: R.: Thank you for your valuable feedback. We have now formatted the specie name in italics as recommended. We appreciate your careful review and suggestions.

Round 2

Reviewer 1 Report

Comments and Suggestions for Authors

I have no further comments on this manuscript.